# Interaction-driven phase transition in one dimensional mirror-symmetry protected topological insulator

**Devendra Singh Bhakuni[1], Amrita Ghosh[1,2] and Eytan Grosfeld[1]**

**1** Department of Physics, Ben-Gurion University of the Negev, Beer-Sheva 8410501, Israel
**2** Physics Division, National Center for Theoretical Sciences, Taipei 10617, Taiwan

## Abstract

Topological crystalline insulators are phases of matter where the crystalline symmetries solely protect the topology. In this work, we explore the effect of many-body interactions in a subclass of topological crystalline insulators, namely the mirror-symmetry protected topological crystalline insulator. Employing a prototypical mirror-symmetric quasi-one-dimensional model, we demonstrate the emergence of a mirror-symmetry protected topological phase and its robustness in the presence of short-range interactions. When longer-range interactions are introduced, we find an interaction-induced topological phase transition between the mirror-symmetry protected topological order and a trivial charge density wave. The results are obtained using density-matrix renormalization group and quantum Monte Carlo simulations in applicable limits.

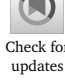

# 1  Introduction

Topological phases of matter have gained enormous interest in condensed matter since their discovery [1,2]. In particular, topological insulators (TIs) of non-interacting fermions are bulk insulators that, in contrast to their conventional counterparts, admit protected gapless surface states. These surface states are protected by the underlying symmetry of the system, and are robust against any perturbations as long as the symmetries are respected.

Symmetry-protected topological states are mainly categorized into two groups based on the symmetry type: non-spatial symmetries and spatial symmetries. Depending on the non-spatial internal symmetries of the system, such as time-reversal, particle-hole and chiral symmetry, topological phases of non-interacting fermions are classified in a 10-fold symmetry class [3–6]. Apart from this conventional classification, recent studies have shown that lattice symmetries also play a crucial role and can be solely responsible for the protection of certain topological phases. These topological phases, arising from spatial symmetries of the system, are called topological crystalline phases [7–13]. One important member of this spatial symmetry group is mirror-symmetry, which has been demonstrated to give rise to various novel phases [11,14–24].

While the search for new topological phases is at the forefront of condensed matter research, the investigation of topological crystalline phases and their robustness against various perturbations and many-body interactions are in a nascent stage. A key question that arises in such systems is how the presence of the many-body interactions affects the topological phases *solely protected* by the crystalline symmetries and, in particular, in subclasses where only mirror-symmetry protects the topological phase.

In this work, we propose a quasi-one-dimensional model of spinless fermions on a zigzag ladder subjected to a staggered on-site potential along both of its legs (Fig. 1(a)). Its one-dimensional embedding (Fig. 1(b)) admits only time-reversal and mirror symmetries placing it in the class we are interested in here. We study the properties of this model in the absence and presence of multi-range interactions and derive its phase diagram.

In the non-interacting limit, the model gives rise to a rich phase diagram consisting of three insulating phases: a charge density wave (CDW) at half-filling and two dimer insulators (DIs) at quarter and three-quarter fillings. The Berry-phase topological invariant as well as the bipartite entanglement entropy identify one of the two DIs as a mirror-symmetry protected TI, depending on the sign of the on-site potential. We obtain the projected Hamiltonians in the

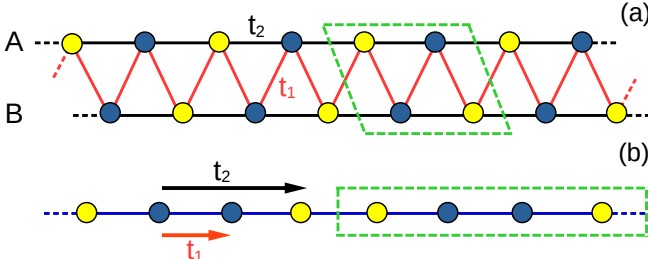

Figure 1: (a): Pictorial description of the zigzag ladder considered in the main text. The indices A and B stand for the two legs of the ladder. The red and black lines represent the NN (inter-leg) and NNN (intra-leg) bonds with hopping strength $t_1$ and $t_2$, respectively. The dashed lines indicate the bonds which connect the ladder across the boundary. Lattice sites denoted by yellow (blue) circles admit on-site potential $-W$ ($W$). (b): Mapping of the zigzag ladder to a 1D chain. The unit cells in both lattices are highlighted by green dashed lines.

two DIs separately which demonstrate an emergent chiral symmetry. This symmetry pins the edge states in the middle of the respective bands also when next-nearest-neighbor (NNN) tunneling ($t_2$) is present. Next, using a combination of exact diagonalization and density-matrix renormalization group (DMRG) approach, we find that the topological phases are remarkably robust against any amount of nearest-neighbor (NN) as well as NNN interactions. Interestingly, the addition of next-to-next nearest-neighbor (NNNN) repulsion gives rise to a phase transition from the topological dimer insulating phase to a topologically trivial CDW phase. The results are also supported by quantum Monte Carlo (QMC) simulations in the limit of zero NNN hopping, where spinless fermions can be mapped into hard-core bosons.

The paper is organized as follows. In Sec. 2 we describe the model and introduce the various tools we employ to characterize its topological phases. In Sec. 3 we detail our characterization considering various settings with zero and non-zero NNN hopping and in the presence of long-range interactions. In Sec. 4 we conclude. In App. A we report QMC simulations.

## 2   Model Hamiltonian and Topological characterization

We consider spinless fermions hopping in a zigzag ladder with staggered on-site potential $W$ applied along its legs $A$ and $B$ as depicted in Fig. 1. Each leg of the ladder consists of $N$ sites. The Hamiltonian of the system can be written as,

$$
\begin{aligned}
H = & -t_1 \sum_i \left( a_i^\dagger b_i + a_i^\dagger b_{i+1} + \text{H.c.} \right) \\
& - t_2 \sum_i \left( a_i^\dagger a_{i+1} + b_i^\dagger b_{i+1} + \text{H.c.} \right) \\
& - W \sum_i (-1)^i n_i^A + W \sum_i (-1)^i n_i^B .
\end{aligned}
\tag{1}
$$

Here $a_i^\dagger$ ($a_i$) is the creation (annihilation) operator of a fermion at site $i$ in leg A and $b_i^\dagger$ ($b_i$) represents the same in leg B. The operator $n_i^A = a_i^\dagger a_i$ ($n_i^B = b_i^\dagger b_i$) denotes the number operator at site $i$ in leg A (B), and $W$ represents the strength of the on-site potential. The NN (inter-leg) hopping is given by $t_1$, while $t_2$ denotes the NNN (intra-leg) hopping.

Under periodic boundary condition (PBC), the Hamiltonian can be written in momentum space as

$$
\mathcal{H}(k) = \begin{bmatrix}
-W & -t_1 & -t_2 \alpha^*(k) & -t_1 \beta^*(k) \\
-t_1 & W & -t_1 & -t_2 \alpha^*(k) \\
-t_2 \alpha(k) & -t_1 & W & -t_1 \\
-t_1 \beta(k) & -t_2 \alpha(k) & -t_1 & -W
\end{bmatrix},
\tag{2}
$$

where $\alpha(k) = 1 + e^{ik}$ and $\beta(k) = e^{ik}$. The Hamiltonian $\mathcal{H}(k)$ is symmetric under the time-reversal operation and also possesses mirror symmetry. Under the time-reversal operation, with $T$ being the anti-unitary time-reversal operator, the Hamiltonian in Eq. 2 behaves as $T\mathcal{H}(k)T^{-1} = \mathcal{H}(-k)$. In this case, the time-reversal operator is simply the complex conjugation operator $\mathcal{K}$, i.e., $T = \mathcal{K}$, which satisfies $T^2 = 1$. On the other hand, under the mirror symmetry, the Hamiltonian obeys: $M\mathcal{H}(k)M^{-1} = \mathcal{H}(-k)$ where $M = \sigma_x \otimes \sigma_x$ is the mirror symmetry operator. Since the two symmetry operators $T$ and $M$ commute with each other: $[T, M] = 0$, the model is a member of the mirror symmetry class AI [7], which admits a $\mathbb{Z}$ topological number in 1D.

The topological characterization can be done using the Berry phase associated with the

Bloch Hamiltonian (Eq. 2) and it is related to the charge polarization $P$ as

$$P = \frac{1}{2\pi} \int_{-\pi}^{\pi} dk \mathcal{A}(k), \tag{3}$$

where $\mathcal{A}(k) = i \sum_n \langle u_{k,n} | \partial_k | u_{k,n} \rangle$ is the Berry's connection and $u_{k,n}$ are the Bloch states at momentum $k$ and band index $n$. Generally, the Berry phase can take any value in between 0 and $2\pi$ (mod $2\pi$), however, for one-dimensional systems exhibiting the time-reversal symmetry and mirror symmetry, the Berry phase provides a $\mathbb{Z}_2$ invariant which is quantized to either 0 or $\pm\pi$ [22, 25].

For the numerical purpose, the Berry phase can be calculated using the Fukui-Hatsugai-Suzuki algorithm [26] which provides a discrete version of the Berry phase as

$$\nu = \frac{1}{\pi} \text{Im} \left[ \log \left( \prod_{i=1}^{N_c} \frac{|U^{(i)}|}{\sqrt{|U^{(i)}||U^{(i)}|^*}} \right) \right], \tag{4}$$

where the Brillouin zone is discretized in $N_c = N/2$ unit cells with lattice points $k_i = -\pi + 2\pi i/N_c$ with $i = 1, 2, \cdots, N_c$ and we have $k_{N_c+1} \equiv k_1$. The elements of $U^{(i)}$ are calculated as $U_{mn}^{(i)} \equiv \langle \psi_m(k_{i+1}) | \psi_n(k_i) \rangle$, with $|\psi_m(k)\rangle$ being the single-particle eigenvectors of $\mathcal{H}(k)$ participating in the relevant filling, and $|U^{(i)}|$ denotes the determinant of $U^{(i)}$.

In addition to the Berry phase, we characterize the phases using the bipartite entanglement entropy. The entanglement entropy quantifies certain correlation between two subsystems of a composite system A∪B. It is defined as $S_A = -\text{Tr}_B(\rho \log \rho)$, where $\rho$ denotes the density matrix of A∪B and $\text{Tr}_B$ represents the partial trace over the subsystem B. The entanglement entropy has been extensively used to characterize various underlying features of both interacting and non-interacting systems including the characterization of quantum phase transitions, topological phase transitions, localization-to-delocalization transitions, and many more [27–34].

For non-interacting particles, a computationally efficient method to calculate the entanglement entropy was demonstrated in Ref. [35]. The method reduces the complexity of the problem by bringing down an exponential dependence on the system size to a polynomial dependence. The procedure constitutes of diagonalization of a two-point correlation matrix, $C_{mn} \equiv \langle c_m^\dagger c_n \rangle$, which inherently takes care of the filling fraction. Here, $c_m^\dagger (c_m)$ creates (annihilates) a fermion at site $m$ in the system. The entanglement entropy is then calculated from the eigenvalues of the correlation matrix $n_i$ as

$$S = -\sum_i [n_i \ln n_i + (1 - n_i) \ln(1 - n_i)]. \tag{5}$$

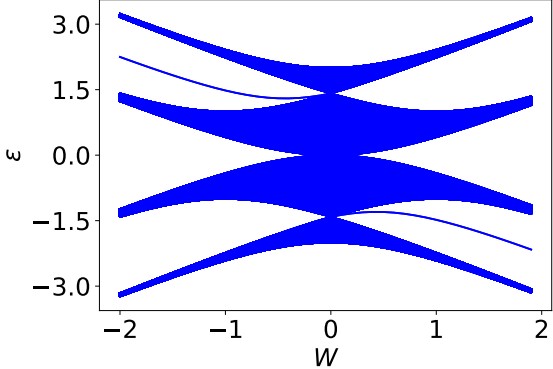

Figure 2: Energy spectrum with OBC with varying values of $W$. The calculations are performed on a ladder with $N = 2000$ sites in each leg, with $t_1 = 1.0$ and $t_2 = 0.0$.

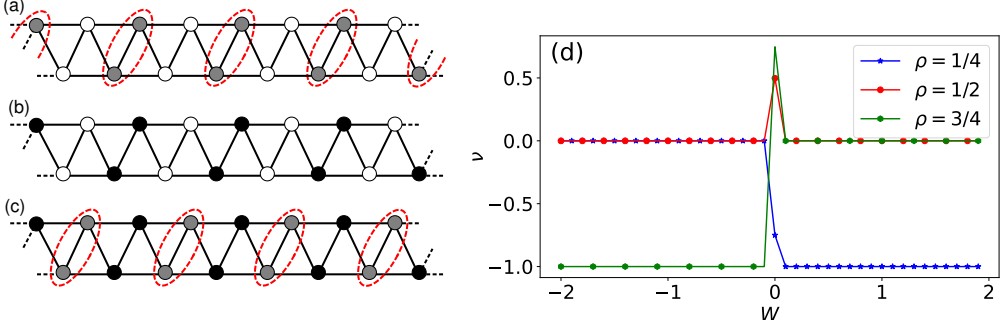

Figure 3: Structures of the three insulating phases of Fig. 2: (a) the dimer insulator at $\rho = 1/4$; (b) the CDW at half-filling; and, (c) the dimer insulator at density 3/4. The white (black) circles denote empty (filled) sites with density 0.0(1.0), whereas the grey circles signify half-filled sites with density 0.5. The red dashed curves represent the formation of dimers. (d) The corresponding Berry phase as a function of $W$ for filling fractions 1/4, 1/2 and 3/4 with $t_1 = 1.0$ and $t_2 = 0.0$.

While this formulation is limited to the non-interacting case, we use DMRG to study the topological properties in the presence of many-body interactions. The numerical simulations for the interacting case are performed using the TeNPy Library [36].

## 3 Results and discussion

### 3.1 Zero NNN hopping

We first consider the Hamiltonian in Eq. 1 without the NNN coupling by setting $t_2 = 0$. The energy spectrum as a function of $W$ under the open boundary condition (OBC) is shown in Fig. 2. The spectrum mainly consists of three gapped phases, corresponding to densities 1/4, 1/2, and 3/4. Interestingly, as the sign of $W$ is changed from negative to positive, an edge state appears in the middle of the energy-gap at density $\rho = 1/4$ indicating a trivial-to-topological phase transition at $W = 0$. This scenario is reversed for the insulator at density 3/4, where a topological-to-trivial phase transition occurs on changing the sign of the potential strength $W$. However, the insulator at half-filling remains trivial throughout.

In the absence of NNN hopping, the system of spinless fermions can be mapped onto a system of hard-core bosons (HCBs). In order to characterize the three above-mentioned insulating phases and investigate their underlying structures, we employ Stochastic Series Expansion (SSE) QMC [37, 38] on the HCB system and study the following order parameters as defined in App. A: the average HCB density $\rho$; the structure factor $S(Q)$; and, the dimer structure factor $S_D(Q)$. The results obtained from QMC calculations, as detailed in App. A, disclose Fig. 3(a),(b) and (c) as the underlying structure of the three insulating phases and the origin of these structures can be understood in the following manner.

In the presence of a staggered potential, half of the lattice sites in the system (depicted by yellow circles in Fig. 1) have lower on-site potential ($-W$) compared to the other half. Therefore, upto half-filling the particles prefer to occupy the lower-potential sites. At 1/4-filling, it is energetically favorable for the system to form dimers occupying the two sites in each red dashed curve in Fig. 3(a), so that the particle can further lower the energy of the system by hopping back and forth between these two sites. For a finite system, this leads to two edge states at $\rho = 1/4$.

For the case of half-filling, all sites with on-site potential $-W$ are completely filled, resulting in a CDW structure as shown in Fig. 3(b). Finally, at 3/4-filling the sites with lower on-site

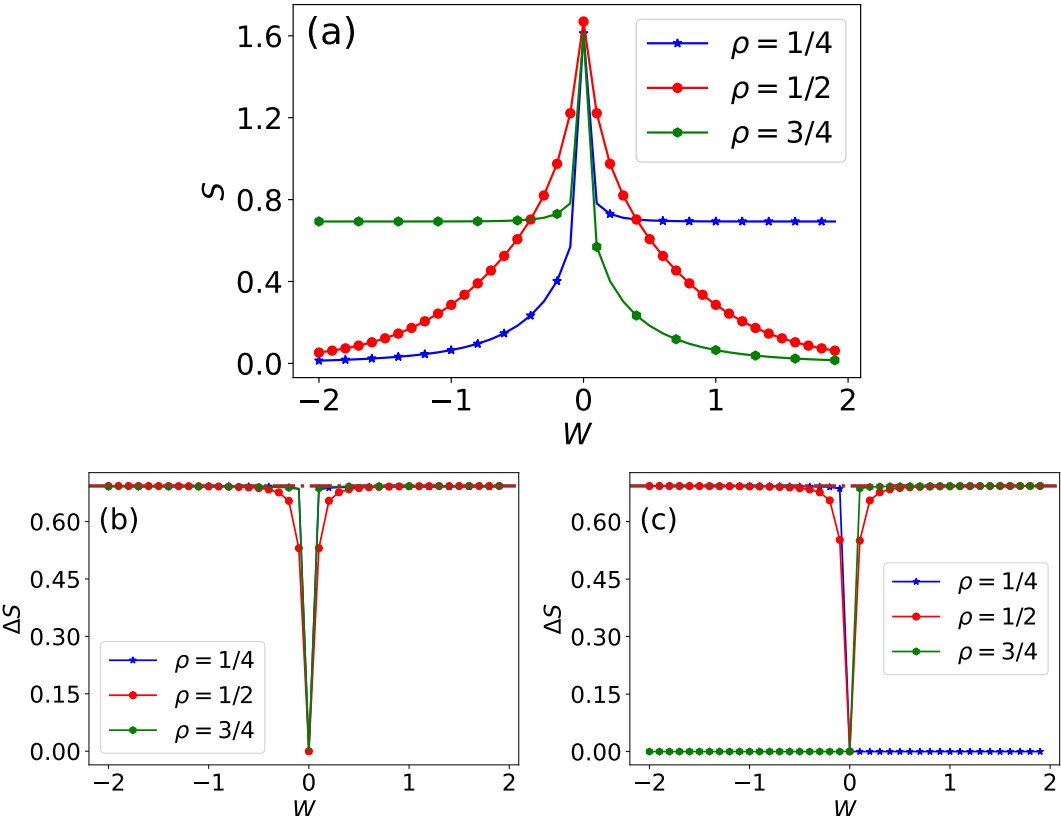

Figure 4: (a) Entanglement entropy as a function of $W$ for filling fractions 1/4, 1/2 and 3/4 with OBC. (b,c) $\Delta S$ as a function of $W$ for PBC and OBC respectively. The other parameters are: $N = 2000$, $t_1 = 1.0$, and $t_2 = 0.0$.

potential are completely filled and the rest are half-filled. Now the particles at these half-filled sites hop back and forth between the two NN sites inside each red curve and form dimers as depicted in Fig. 3(c). It is important to note that there is a major difference between the structures of the two DIs. In case of the dimer insulator at 1/4-filling there exists a dimer which involves two edge sites under OBC. Consequently, for $W > 0$ this dimer insulator displays the existence of edge states as depicted in Fig. 2. However, for the case of $\rho = 3/4$, the dimers are entirely formed in the bulk and the edge state does not appear when OBC is applied. We should note that when the sign of the on-site potential is reversed, the appearance of edge states is also changed. In this situation, the dimer insulator at $\rho = 1/4$ contains bulk dimers only, whereas the insulator at 3/4-filling shows edge states.

The topological invariants corresponding to densities 1/4, 1/2 and 3/4, are shown in Fig. 3(d) as a function of $W$. One can see that for the insulator at $\rho = 1/4$, the Berry phase is quantized at $-1$ for positive values of $W$, whereas for negative values of $W$, it remains zero. The situation is reversed for the insulator at density 3/4. In this case, the Berry phase remains zero for all positive values of $W$ and quantized at $-1$ for negative $W$ values. Thus we can identify the DIs at $\rho = 1/4$ and $\rho = 3/4$ as TIs for $W > 0$ and $W < 0$, respectively. On the other hand, for the insulator at $\rho = 1/2$, the Berry phase remains zero throughout for all values of $W$, which makes it topologically trivial.

The presence of these distinct phases and the edge states can also be characterized by studying the bipartite entanglement entropy of the ground state. We consider the left and right parts of the ladder, constituting an equal number of sites, as two sub-systems. The entanglement entropy for different filling fractions with OBC is plotted in Fig. 4(a). For $\rho = 1/4$, the

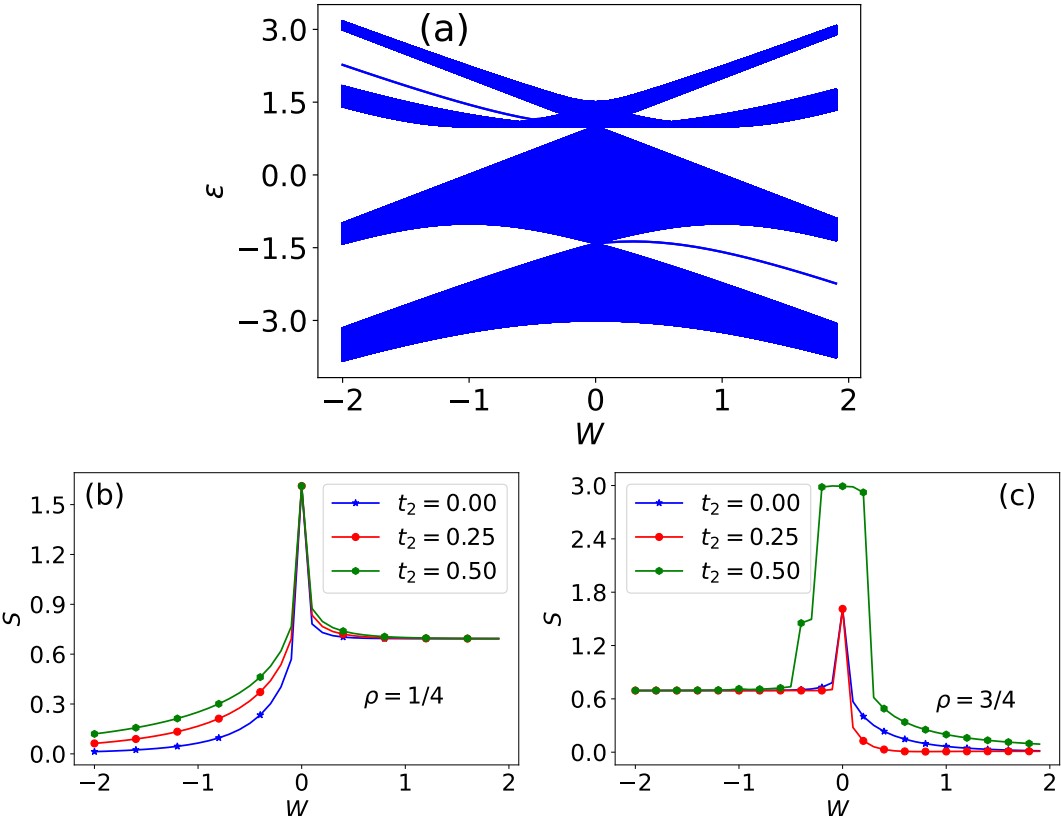

Figure 5: (a) Energy spectrum as a function of $W$ for a non-zero NNN hopping $t_2 = 0.5$. (b,c) Entanglement entropy as a function of $W$ for different values of NNN hopping $t_2$ with OBC: (b) for $\rho = 1/2$; (c) for $\rho = 3/4$. Here $N = 2000$ and $t_1 = 1.0$.

entanglement entropy increases with increasing $W$ indicating a signature of quantum phase transition at $W = 0$, after which it saturates to $\ln 2$ in the topological phase. A reverse trend can be seen for $\rho = 3/4$ where the topological phase exists for $W < 0$. On the other hand, for $\rho = 1/2$, we see a symmetric pattern. To observe the signature of edge state, we calculate the change in the entanglement entropy when an additional particle is either added or removed. We define the quantity $\Delta S_\rho^{(\pm 1)}$ as [34]

$$\Delta S_\rho^{(\pm 1)} = S_\rho^{(\pm 1)} - S_\rho, \qquad (6)$$

where $S_\rho$ is the entanglement entropy at filling fraction $\rho$ and $S_\rho^{(\pm 1)}$ is the same but with a single particle either added or removed. For our calculation, the difference $\Delta S \equiv \Delta S_\rho^{(-1)}$ is calculated by considering the entanglement entropy at $\rho = 1/4$, $1/2$, and $3/4$, and then removing a single particle from these filling fractions. The behavior of $\Delta S$ as function of $W$ is plotted in Fig. 4(b,c). While under PBC (Fig. 4(b)) the behavior remains the same for all the filling fractions, the case of OBC (Fig. 4(c)) shows similar behavior as that of the Berry phase (Fig. 3(d)). Specifically, for $\rho = 1/4$, we see a transition from $\Delta S = \ln 2$ at $W < 0$ to $\Delta S = 0$ at $W > 0$; and, for $\rho = 3/4$, the trend reverses and a transition from $\Delta S = 0$ to $\Delta S = \ln 2$ is observed. These transitions are taken to signify the presence of edge states for these filling fractions. In contrast, for $\rho = 1/2$, the behavior of $\Delta S$ is similar for OBC and PBC, suggesting a trivial phase at this filling fraction.

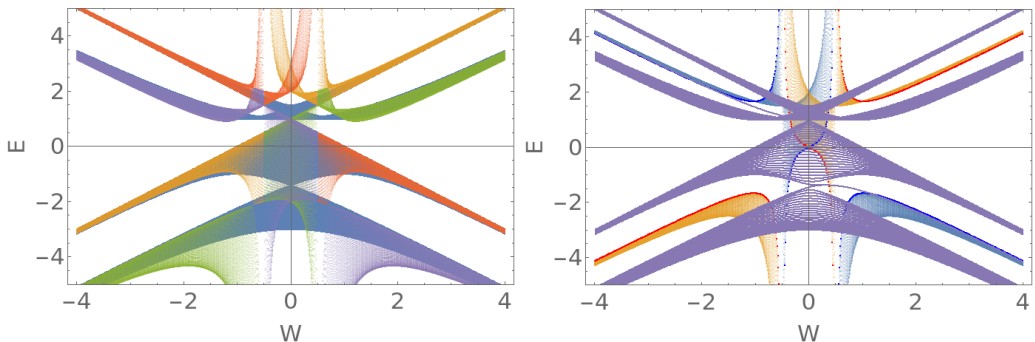

Figure 6: Comparison between the exact spectrum and the spectrum obtained from the projected Hamiltonian. Left panel: the exact spectrum for PBC (blue) and for the lower and upper projected two bands (red, orange, green, purple) as function of $W$. Right panel: the spectrum for OBC (purple) and the dispersing centers $\varepsilon_0(k)$ (blue) and $\varepsilon_0'(k)$ (orange) with $k = \pi$ highlighted (bright blue and red respectively). Here $t_1 = 1.0$, $t_2 = 0.5$, and $N = 50$.

## 3.2 Non-zero NNN hopping

We now consider the case of non-zero NNN hopping. The energy spectrum for $t_1 = 1.0$ with $t_2 = 0.5$ as a function of $W$ is plotted in Fig. 5(a). It can be seen that while the edge state for the filling $\rho = 1/4$ remains almost unchanged in the presence of a non-zero $t_2$, a gapless phase appears at 3/4-filling for small values of $W$. The same observation can also be seen from the entanglement entropy plotted in Fig. 5(b,c) for different values of $t_2$. For the case of $\rho = 1/4$, we see that the behavior of the entanglement entropy is qualitatively similar for different values of $t_2$ (Fig. 5(b)) which suggests that the topological phase in this filling remains unaffected by $t_2$. For $\rho = 3/4$, the quantization of the entanglement entropy and appearance of the topological phase shifts to larger values of $|W|$ with increasing $t_2$ (Fig. 5(c)). The topological phase, however, remains stable for large enough value of the staggered potential $W$. The robustness of these topological phases at $\rho = 1/4$ and $\rho = 3/4$ can be explained by the presence of an emergent symmetry.

While the Hamiltonian in Eq. (1) admits a mirror symmetry which protects the topological phases, there is no chiral symmetry in the system. However one can show that there is indeed an emergent chiral symmetry, for the lower and upper bands separately, which pins the edge states to the center of the respective bands. By a projection to the lower two bands of the system for $W > 0$ (or the upper ones for $W < 0$), we can write an effective Hamiltonian

$$\hat{H}_{\text{proj}} = \varepsilon_0(k)\tau_0 + \varepsilon_x(k)\tau_x + \varepsilon_y(k)\tau_y, \tag{7}$$

with

$$\varepsilon_0(k) = -W - \frac{2t_1^2 W + 2t_2\left(t_1^2 + 2t_2 W\right)(1 + \cos k)}{4W^2 - t_1^2}, \tag{8}$$

$$\varepsilon_x(k) = -t_1\left[\cos k + \frac{t_1^2 + 2t_2 \cos k\left(t_2(1 + \cos k) + 2W\right)}{4W^2 - t_1^2}\right], \tag{9}$$

$$\varepsilon_y(k) = -t_1\left[\sin k + \frac{2t_2 \sin k\left(t_2(1 + \cos k) + 2W\right)}{4W^2 - t_1^2}\right]. \tag{10}$$

In Eq. (7) $\tau_0$ is the $2 \times 2$ identity matrix and $\tau_i$ ($i = x, y, z$) is the $i$'th Pauli matrix. The resulting bands are plotted in Fig. 6 (left panel, purple and red), matching quite well with the

exact spectrum for larger values of $|W|$. The projected Hamiltonian of the lower two bands reveals a partial chiral symmetry $\tau_z$. Moreover, it has a topological number for both positive and negative larger values of $W$. Consequently, it admits an edge state which gets pinned at the middle of the band gap at 1/4-filling for positive $W$ and at 3/4-filling for negative $W$. While the center of the band is dispersing with energy $\varepsilon_0(k)$, the edge state gets locked to $k = \pi$ as demonstrated in Fig. 6, right panel.

Performing a similar projection to the upper two bands for $W > 0$ (or the lower ones for $W < 0$), we get the projected Hamiltonian

$$\hat{H}'_{\text{proj}} = \varepsilon'_0(k)\tau_0 + \varepsilon'_x(k)\tau_x + \varepsilon'_y(k)\tau_y \,, \tag{11}$$

where the expressions of $\varepsilon'_0(k)$, $\varepsilon'_x(k)$ and $\varepsilon'_y(k)$ in this case reduce to,

$$\varepsilon'_0(k) = W + \frac{\left(2t_1^2 W + 2t_2\left(2t_2 W - t_1^2\right)(1 + \cos k)\right)}{4W^2 - t_1^2} \,, \tag{12}$$

$$\varepsilon'_x(k) = -t_1\left[1 + \frac{\left(t_1^2 \cos k + 2t_2\left(t_2 - 2W\right)(1 + \cos k)\right)}{4W^2 - t_1^2}\right] \,, \tag{13}$$

$$\varepsilon'_y(k) = -t_1\frac{\left(t_1^2 - 4t_2 W\right)\sin k}{4W^2 - t_1^2} \,. \tag{14}$$

The resulting bands are plotted in Fig. 6 (left panel, green and orange). Therefore the projected Hamiltonian for the upper two bands also admits a chiral symmetry $\tau_z$. In contrast to the previous case, this Hamiltonian has no topological number in its regions of validity.

### 3.3 Effect of interactions

We now study the stability of the TIs in the presence of many-body interactions. To demonstrate the effect of interactions we choose the TI at density 1/4. We have confirmed that the TI at $\rho = 3/4$ also behaves exactly the same way. We consider a NN repulsion between the fermions by adding the following term to the Hamiltonian in Eq. (1),

$$H_1 = V_1 \sum_i (n_i^A n_i^B + n_i^A n_{i-1}^B) \,. \tag{15}$$

Fig. 7(a) shows the variation of the entanglement entropy $S$ as we tune the NN repulsion strength $V_1$. We consider various different sets of $(W, t_2)$ values. As mentioned previously, for $V_1 = 0$, the insulator at $\rho = 1/4$ is topological (non-topological) in nature for $W > 0$ ($W < 0$), for both zero and nonzero values of $t_2$. We see that in the presence of NN repulsion, the topological nature of the insulator at $\rho = 1/4$ remains unaltered as $S$ remains quantized at $\ln 2$ for all values of $V_1$ (Fig. 7(a)). In contrast, for $W < 0$, the topologically trivial insulator remains the same as a function of increasing $V_1$ with a vanishingly small value of the entanglement entropy. Fig. 7(c) represents the variation of Zak phase calculated using the twisted boundary conditions [39–42] as a function of increasing NN repulsion $V_1$, measured on a ladder with $N = 8$ for the same sets of $(W, t_2)$ values using exact diagonalization. The quantization of the topological invariant at 1(0) for the positive(negative) value of the on-site potential $W$ essentially supports the entanglement entropy results. In fact, one can argue that the topological nature of the insulator at $\rho = 1/4$ will be protected against any amount of NN repulsion. Since the dimers in the $\rho = 1/4$ dimer insulator are formed at every fourth NN bond, the particles forming two neighboring dimers do not feel any repulsion among them. As a result, the NN repulsion does not interrupt the hopping process necessary to form dimers. Thus, the

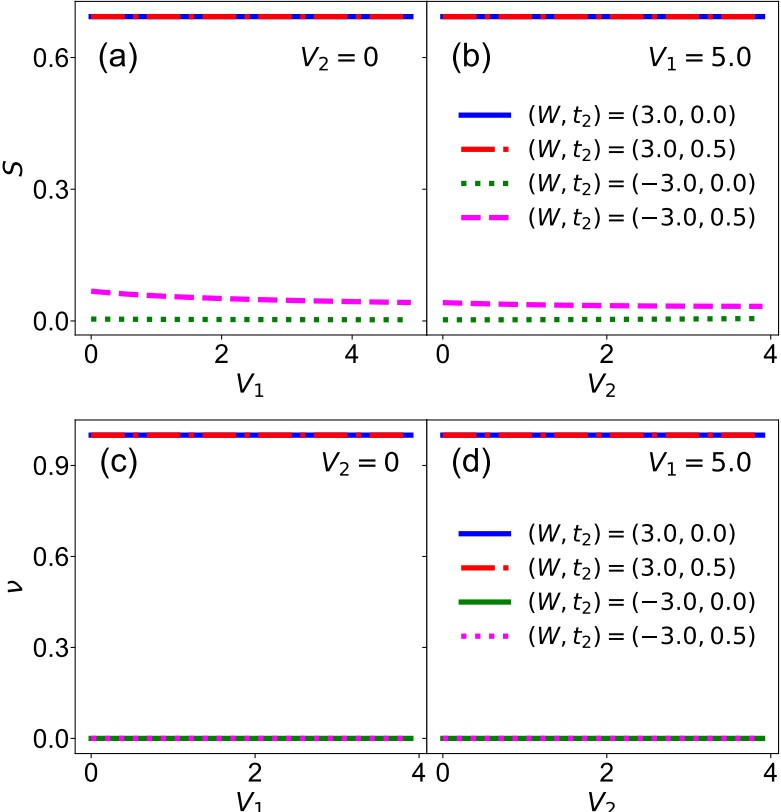

Figure 7: Entanglement entropy $S$ as a function of : (a) NN repulsion $V_1$ with $V_2 = 0$; and, (b) NNN repulsion $V_2$ with $V_1 = 5.0$, both measured under OBC. The measurements are performed for four different sets of $(W, t_2)$ values detailed in the legend with $N = 400$. Zak phase as a function of (c) NN repulsion $V_1$ with $V_2 = 0$; and, (d) NNN repulsion $V_2$ with $V_1 = 5.0$, measured for a ladder with $N = 8$ for four different sets of $(W, t_2)$ values.

topological nature of the dimer insulator remains intact and unaffected in the presence of any amount of NN repulsion.

We next consider the effect of NNN repulsion by adding another term in the Hamiltonian, given by,

$$H_2 = V_2 \sum_i (n_i^A n_{i+1}^A + n_i^B n_{i+1}^B). \tag{16}$$

In Fig. 7(b) we show the variation of the entanglement entropy $S$ as a function of the NNN repulsion strength $V_2$ for a fixed value of NN repulsion, $V_1 = 5$. We see that the effect of NNN repulsion on the dimer insulator is similar to the effect of NN repulsion. Since the entanglement entropy remains quantized at $\ln 2$ with varying $V_2$, the topological nature of the insulator is robust against the NNN repulsion. The same feature emerges from the variation of topological invariant as well (see Fig. 7(d)). As the dimers are formed on every fourth NN bond, the fermions on two neighboring dimers are shielded from the NNN repulsion. Therefore, similarly to the case of NN repulsion, the TI is robust against any amount of NNN repulsion as well.

Finally, we explore the stability of the topological phase in the presence of third nearest-neighbor or next-to-next nearest-neighbor (NNNN) repulsion, by adding the following term to

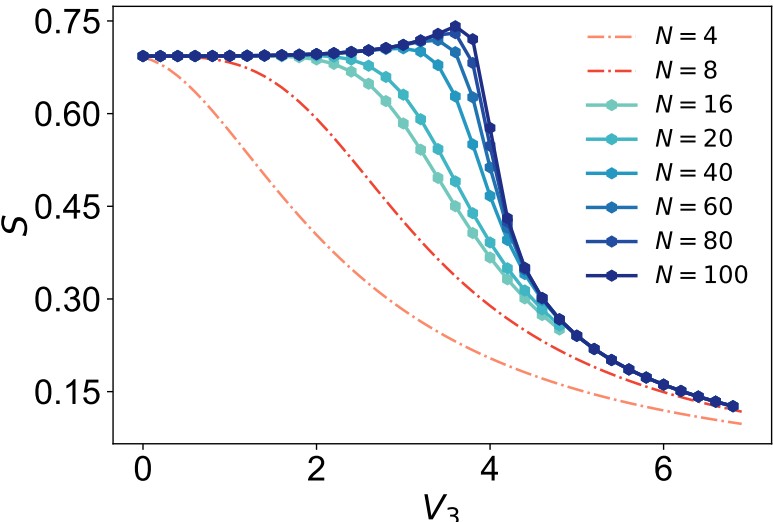

Figure 8: Entanglement entropy $S$ as a function of NNNN repulsion $V_3$ with $V_1 = 5.0, V_2 = 4.0, W = 3.0, t_1 = 1.0$ and $t_2 = 0.0$, measured under OBC.

the Hamiltonian,

$$H_3 = V_3 \sum_i (n_i^A n_{i+1}^B + n_i^A n_{i-2}^B). \tag{17}$$

The entanglement entropy in the OBC as a function of $V_3$ is shown in Fig. 8, for different system sizes using exact diagonalization (dashed dotted line), as well as DMRG (solid lines). We can see that a sufficiently large value of $V_3$ destroys the topological phase, where the entanglement entropy goes from a quantized value of ln 2 towards zero. This result can be understood by realizing that unlike the NN and NNN repulsions, the dimers formed at every fourth NN bond do feel the presence of a NNNN repulsion. As a result the hopping back and forth of a particle in a dimer gets interrupted. In order to avoid repulsion, for a sufficiently large value of $V_3$, it becomes energetically favorable for the particles to just occupy all sites with negative $W$ in either leg A or leg B of the ladder. This leads to a topologically trivial CDW phase. The emergence of this CDW phase is further supported by the QMC analysis (App. A) where the dimer structure factor and the structure factor undergo a transition on varying the strength of the repulsion $V_3$ suggesting a dimer insulator at a small value of $V_3$ and a CDW phase for large enough value of $V_3$.

## 4 Summary and discussion

To summarize, we have considered spinless fermions, with NN as well as NNN hopping, on a zigzag ladder subjected to staggered on-site potential along its two legs. The system reveals the existence of three gapped phases at 1/4, 1/2 and 3/4 filling fractions. The insulator at 1/2-filling turns out to be a CDW in nature, whereas the other two gapped phases are characterized as DIs. One of these two DIs emerges as a TI, depending on the sign of the on-site potential $W$. We have characterized the topological nature of these insulators using the Berry phase as well as entanglement entropy. The topological phase is protected by mirror-symmetry which places the system in the AI mirror-symmetry class. Additionally, performing a projection to the lower and upper two bands separately, we have shown that the system admits an emergent chiral symmetry which pins the edge states of the TIs to the middle of the corresponding energy bands also in the presence of NNN hopping.

Interestingly, since the dimers formed in the system are well-separated, the topological phases become stable in the presence of repulsive interactions of any strength up to NNN. However, introducing longer-range repulsive interactions (NNNN) destroys the topological phase by transforming it into a topologically trivial CDW phase. We believe that our theoretical proposal can be tested in artificially engineered systems, such as cold atoms in optical lattices, which provide a precise control over the tunable parameters of the system and have become a prolific venue for realizing various phases of non-interacting as well as interacting fermions and bosons [43–53]. We note that if our system is realized as a zigzag ladder, rather than its one-dimensional variant, the mirror symmetry should be implemented in the physical system using a confining potential $V(x, y)$ that is inversion-symmetric.

The model discussed here comprises in essence two effective intertwined Su-Shrieffer-Heeger (SSH) models which are separated in energy due to the on-site potential. It should be noted that a single SSH model admits, in addition to mirror symmetry, also a chiral symmetry, which makes it unsuitable for our purpose here. Due to the one-dimensional nature of the system, the topological insulating phases obtained are sometimes referred to as *symmetry-obstructed atomic insulators* and have a very unconventional bulk-boundary correspondence. For example, the presence of a boundary potential can move the edge states to the bulk without changing the topology of the system [22]. Nevertheless, the eventual transition from the topological phase into a CDW, driven by strong longer-range interactions, is a bulk effect, as demonstrated by the supporting quantum Monte Carlo simulations.

## Acknowledgements

This research was funded by the Israel Innovation Authority under the Kamin program as part of the QuantERA project InterPol, and by the Israel Science Foundation under grant 1626/16. DSB and AG thank the Kreitman School of Advanced Graduate Studies for support. AG would also like to thank Ministry of Science and Technology, National Center for Theoretical Sciences of Taiwan for support towards the end of this project.

## A    QMC calculations

In this section we detail the QMC calculations and results obtained therefrom, for a system of HCBs obeying the Hamiltonian given in Eq. (1), in the absence of NNN hopping. First, we describe the three order parameters used in QMC calculations, namely: the average HCB density $\rho$, the structure factor $S(Q)$, and the dimer structure factor $S_D(Q)$.

The average density of a system containing $N_s$ sites can be calculated as

$$\rho = \frac{1}{N_s} \sum_i n_i \,, \tag{18}$$

where $n_i$ gives the number of HCBs ( 0 or 1) at site $i$.

The structure factor per site can be calculated as,

$$S(Q) = \frac{1}{N_s^2} \sum_{i,j} e^{iQ(r_i - r_j)} \langle n_i n_j \rangle \,, \tag{19}$$

where $\langle \cdots \rangle$ represents ensemble average and $r_i$ denotes the position of site $i$. The zigzag ladder can always be represented as a one-dimensional chain by straightening the red bonds (Fig. 1(b) in the main text). In the above expression for the structure factor we use the position vectors of this transformed 1D chain and its corresponding momentum values as $Q$.

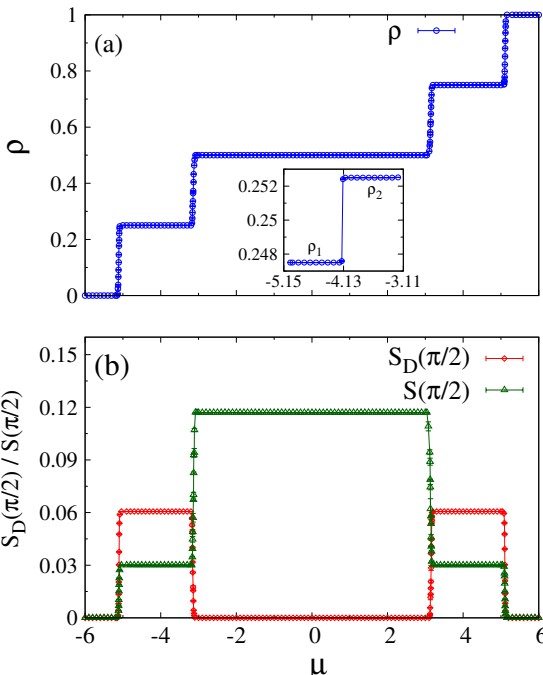

Figure 9: Variations of the order parameters: (a) HCB density $\rho$ and (b) structure factor $S(\frac{\pi}{2})$ and dimer structure factor $S_D(\frac{\pi}{2})$, as a function of the chemical potential $\mu$. Inset of (a): Splitting of $\rho = 1/4$ plateau under OBC. The measurements are done on a ladder with $N = 200$, where $t_1 = 1.0$, $t_2 = 0.0$ and $W = 4.0$.

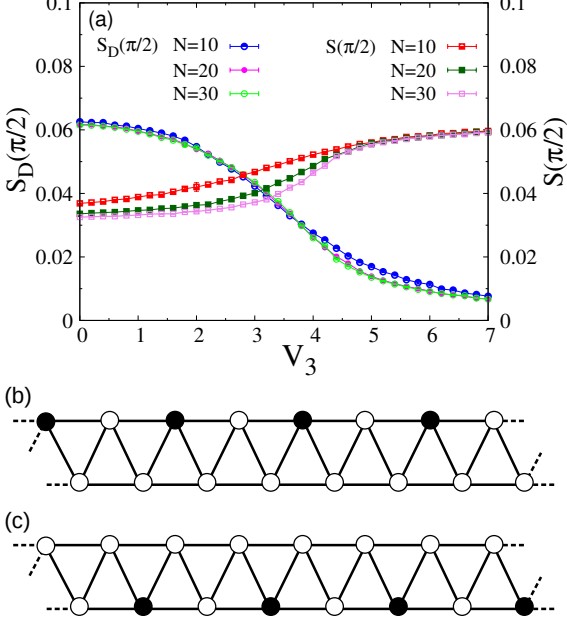

Figure 10: (a) Variations of dimer structure factor $S_D(\frac{\pi}{2})$ and structure factor $S(\frac{\pi}{2})$, as a function of $V_3$. The measurements are done for three different sizes of ladder with $t_1 = 1.0$, $t_2 = 0.0$, $W = 3.0$, $V_1 = 5.0$ and $V_2 = 4.0$. (b-c) Schematic diagram of two possible structures of the CDW at density $\rho = 1/4$ occurring for large values of $V_3$.

Next, the dimer structure factor is defined as

$$S_D(Q) = \frac{1}{N_b^2} \sum_{\alpha,\beta} e^{iQ(R_\alpha - R_\beta)} \langle D_\alpha D_\beta \rangle, \tag{20}$$

where $R_\alpha$ denote the midpoints of the NN bonds of the transformed 1D chain and the dimer operator $D_\alpha = d_{\alpha_L}^\dagger d_{\alpha_R} + d_{\alpha_R}^\dagger d_{\alpha_L}$ is defined on the $\alpha$-th NN bond. Here $\alpha_L$ and $\alpha_R$ represent the two lattice sites attached to this bond and, $d_{\alpha_L}^\dagger$ ($d_{\alpha_L}$) and $d_{\alpha_R}^\dagger$ ($d_{\alpha_R}$) create(annihilate) a HCB at these two sites, respectively. The summation in the above expression runs over the NN bonds in order to detect the formation of dimers along these bonds only.

Fig. 9 displays the variations of the order parameters, namely the HCB density $\rho$, structure factor $S(\pi/2)$ and dimer structure factor $S_D(\pi/2)$ as a function of the chemical potential $\mu$ for a fixed value of on-site potential strength $W = 4$. Here we have set the NN and NNN hopping to be $t_1 = 1.0$ and $t_2 = 0.0$, respectively. It is clear from Fig. 9(a) that in the absence of NNN hopping, there exists three incompressible insulating phases corresponding to the three plateaus. To characterize the nature of these insulators we have calculated the dimer structure factor $S_D(Q)$ and structure factor $S(Q)$ for all values of $Q$ and identify $Q = \pi/2$ to be the one at which both of them peak. Fig. 9(b) displays the change in $S_D(\pi/2)$ and $S(\pi/2)$ as we tune the chemical potential of the system. The dimer structure factor shows a peak at densities 1/4 and 3/4 with a value very close to 0.0625, whereas the structure factor attains a value close to 0.125 at half-filling. The structures of the three above-mentioned insulating phases are shown in Fig. 3(a),(b) and (c). In terms of the transformed 1D lattice, both of the DIs consist of dimers at every fourth NN bond. Therefore in Fig. 9 the dimer structure factor peaks at $Q = \pi/2$ with a value 0.0625. On the other hand for the CDW at half-filling the sites in the red dashed curve corresponding to dimers in Fig. 3(a) become completely filled, while the rest of the sites are empty. Therefore, for this structure the dimer structure factor vanishes completely and the structure factor attains the maximum value 0.125 at wavevector $Q = \pi/2$ as depicted in Fig. 9.

In QMC calculations the existence of the edge states is manifested in the following way. For $W > 0$, under OBC the variation of the average HCB density $\rho$ as a function of the chemical potential $\mu$ remains unchanged except for $\rho = 1/4$. For a ladder with $N = 200$ sites in each leg, the plateau at 1/4-filling with PBC splits into two plateaus corresponding to densities $\rho_1 = 0.2475$ and $\rho_2 = 0.2525$ once we open the boundary of the system (inset of Fig. 9(a)). The values of $\rho_1$ and $\rho_2$ depend on the size of the system. For a system with $2N$ total number of sites, the plateau at density $\rho$ splits into $\rho_1 = \rho - 1/(2N)$ and $\rho_2 = \rho + 1/(2N)$, such that $(\rho_2 - \rho_1) \times 2N = 2$ gives the number of edge states in the system. Therefore the splitting of the plateau under OBC proves the existence of the edge states in the system. The lower plateau ($\rho_1$) corresponds to a situation when both of the edge sites are empty, while the upper plateau ($\rho_2$) signifies a situation when both of them are occupied.

Next, we study the effect of NNNN repulsion on the topological dimer insulator phase at density $\rho = 1/4$. Fig. 10(a) depicts the variations of the dimer structure factor $S_D(\pi/2)$ and structure factor $S(\pi/2)$ as a function of NNNN repulsion $V_3$. As $V_3$ is tuned from zero, the dimer structure factor $S_D$ at wave vector $\pi/2$ starts decreasing from 0.0625 towards zero. Concomitantly, the structure factor $S(\pi/2)$ increases from a value close to 0.03125 to a value close to 0.0625. These results can be understood in the following manner. At $V_3 = 0$, at density 1/4 the system is in a topological dimer insulator phase, whose structure is depicted in Fig. 3(a). As discussed earlier in this phase the dimer structure factor $S_D(\pi/2)$ peaks with a value 0.0625, whereas the structure factor $S(\pi/2)$ attains a value very close to 0.03125. Now, as the value of $V_3$ is increased the particles forming the dimers start to feel the repulsion. Consequently beyond some critical value of $V_3$, it becomes energetically favorable for the particles to occupy the negative-potential sites in either leg A or leg B of the ladder. This gives rise to a CDW

phase which has two possible structures as shown in Fig. 10(b,c). For this CDW the structure factor should peak at wavevector $\pi/2$ with a value 0.0625, which can also be observed from our results in Fig. 10(a). An interesting point to note is that, since the dimer insulator phase at $V_3 = 0$ can be thought of as the fluctuation between the two CDW structures (Fig. 10(b) and (c)), the structure factor $S(\pi/2)$ in the dimer insulator phase is exactly half of the value in the CDW phase.

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
