# Peer review of "Interaction-driven phase transition in one dimensional mirror-symmetry protected topological insulator"

_SciPost Physics, doi:SciPost Phys. Core 5, 048 (2022)_

## Round 1 · Referee Report · Anonymous (Referee 1) · 2022-3-31

Report

Bhakuni et al studied the topological phases of a ladder model with four electrons per unit cell. The authors find its phase diagram, corresponding to a CDW phase and two dimerized phases at different fillings, the latter characterized by a quantized Berry phase due to the mirror symmetry in the ladder. The authors see that this result is stable against nearest neighbor and next-nearest neighbor interactions using DMRG and QMC.

In my view, the results are relatively unsurprising, they can be easily argued from a filling anomaly point of view, and the consequences of interactions are underwhelming since there is no qualitative change in the physics once interactions come into play. Nonetheless, I found the careful treatment with respect to the entanglement entropy insightful and I do believe the work is scientifically sound and worthy of publication.

The manuscript would, however, benefit from drawing a comparison to the simple SSH model, and referencing the works that treat the filling anomaly of obstructed atomic phases. Since the protecting symmetry is a mirror (just like in the SSH) there is no necessity for chiral symmetry. This is further apparent in the spectrum around the topological gaps, where the edge mode is not pinned to the center of the gap (Fig 2). In this sense calling it a TI may be a little misleading, and an obstructed atomic limit is possibly a more fitting term.
  • validity: high
  • significance: ok
  • originality: ok
  • clarity: good
  • formatting: good
  • grammar: excellent

Author:  Eytan Grosfeld  on 2022-07-05  [id 2635]

(in reply to Report 1 on 2022-03-31)

We thank the Referee for reading our work and for his/her comments.

Bhakuni et al studied the topological phases of a ladder model with four electrons per unit cell. The authors find its phase diagram, corresponding to a CDW phase and two dimerized phases at different fillings, the latter characterized by a quantized Berry phase due to the mirror symmetry in the ladder. The authors see that this result is stable against nearest neighbor and next-nearest neighbor interactions using DMRG and QMC. In my view, the results are relatively unsurprising, they can be easily argued from a filling anomaly point of view, and the consequences of interactions are underwhelming since there is no qualitative change in the physics once interactions come into play. Nonetheless, I found the careful treatment with respect to the entanglement entropy insightful and I do believe the work is scientifically sound and worthy of publication.

We are glad to learn that the Referee finds the analysis of the entanglement entropy insightful and the manuscript as scientifically sound and worthy of publication. To strengthen our manuscript, we have now added a discussion of the NNNN repulsion to the revised manuscript which shows a transition from a topological non-trivial dimer insulating phase to a topologically trivial charge density wave phase on varying the interaction strength. The phase transition is also exposed using the entanglement entropy (ED, DMRG) and the various order parameters (QMC). We believe our model provides one of the simplest lattice geometries that admits a topological phase protected by a crystalline symmetry which is quasi-one-dimensional hence amenable to study in presence of interactions.

The manuscript would, however, benefit from drawing a comparison to the simple SSH model, and referencing the works that treat the filling anomaly of obstructed atomic phases. Since the protecting symmetry is a mirror (just like in the SSH) there is no necessity for chiral symmetry. This is further apparent in the spectrum around the topological gaps, where the edge mode is not pinned to the center of the gap (Fig 2). In this sense calling it a TI may be a little misleading, and an obstructed atomic limit is possibly a more fitting term.

We have now added a paragraph drawing a comparison with the SSH model. Our model comprises two intertwined SSH models at separated energies. It is protected by mirror symmetry but has no chiral symmetry. However in the effective model we derive in Sec. III B for the two dimer-insulators, even in the presence of finite $t_2$ an emergent chiral symmetry ensures the edge states stay close to the center of the lower two (and upper two) bands at larger values of $|W|$, inherited from the two partial SSH chains. We also added a paragraph to the paper explaining the nomenclature of obstructed atomic limit.

---

## Round 1 · Referee Report · Alexander Lau (Referee 2) · 2022-4-5

Strengths

1 - interesting study of a topological model linking non-interacting topology with interaction physics
2 - the paper has a clear structure and reads nicely
3 - the calculations are sound and the results seem correct overall
4 - the manuscript provides all necessary details to understand the research carried out and the results
5 - provides an intuitive picture based on the formation of dimers to understand the numerical results, in particular the robustness of the topological phases in the presence of interactions

Weaknesses

1 - the protecting symmetry is not a real mirror symmetry in the considered ladder model rendering the protection quite weak
2 - the introduction and discussion of the Berry phase as a topological invariant is flawed

Report

In this manuscript, the authors study a 1D model of spinless fermions on a zigzag ladder giving rise to topological phases protected by mirror symmetry. They identify a staggered on-site potential to be the important ingredient to obtain topological phases. They determine the ground-state character of the emerging phases as charge-density waves and dimer insulators depending on the filling. The topological nature of the phases is established by Berry phase and entanglement entropy calculations. Moreover, an emergent chiral-like symmetry pins the occuring end states to the center of the gap. The authors further analyze the stability of the topological phases towards interactions using DMRG and QMC. They numerically find that the phases are robust against nearest- and next-nearest neighbor repulsions and provide an intuitive explanation for this finding in terms of the dimer character of the corresponding ground states.

This is an interesting study of a topological model linking non-interacting topology with interaction physics. It is quite timely, in my opinion. The paper has a clear structure and reads nicely. The calculations are sound and the results seem correct overall. The manuscript provides all necessary details to understand the research carried out and the results. I particularly appreciate the intuitive explanations based on the formation of dimers to understand the properties of the system.

Nevertheless, I have two major points of criticism:

First of all, the system’s topology is protected by a symmetry that corresponds to a reflection symmetry only in the chain version of the model. In the ladder model, which is the focus of this paper, this symmetry is not a crystalline symmetry, but rather an emergent/artificial symmetry because it doesn’t correspond to an actual mirror plane/line. In other words, the topology in the zigzag ladder is not protected by a real mirror symmetry. This, in turn, renders the protection weaker because the topology is protected “only” by an unphysical symmetry. The authors should be clearer about this and revise their manuscript in this light. In particular, I think the attribute “Mirror-symmetry protected” in the title is misleading in this regard.

Second, the introduction and discussion of the Berry phase as a topological invariant is problematic. Generally, the Berry phase is not an invariant, because it can assume any value between 0 and $2\pi$ (mod $2\pi$). However, it works here because the pseudo-reflection symmetry of the model quantizes the Berry phase to 0 or $\pi$ mod $2\pi$ [see Zak, PRL 62, 2747 (1989) or Lau et al., PRB 94, 165164 (2016)]. This is not discussed in the manuscript but merely a numerical observation. Also, even in this case, the Berry phase does not give access to the full $\mathbb{Z}$ topological invariant but only to a $\mathbb{Z}_2$ factor of it. This part of the paper should be revised and the quantization be discussed explicitly.

Despite my criticism, I think that the paper’s overall results and the type of study are interesting enough to be considered for publication in SciPost Physics after sufficient and satisfactory revision.

Requested changes

1 - The authors should be clearer about the considered reflection symmetry and revise their manuscript in this light (see report). 2 - The part of the paper about the Berry phase invariant should be revised and the quantization be discussed explicitly (see report). 3 - After Eq.(2), I think it would help to define the basis of the Hamiltonian with respect to Fig. 1 and also indicate the corresponding unit cell in Fig. 1. 4 - The notation $\mathbb{Z}$ for the invariant in Eq.(3) is problematic. The symbol $\mathbb{Z}$ is used in the literature to denote the set or group of integer numbers. The topological invariant assumes values in this set. Furthermore, as pointed out above, the Berry phase, if quantized, only gives a $\mathbb{Z}_2$ factor of the full invariant. The authors should rather use a greek or latin character for the invariant. 5 - Could the authors elaborate more on the origin of Eq.(3)? This formula does not explicitly appear in Ref.34. In particular, I do not see why the normalization term in the denominator is needed. In fact, this formula is otherwise identical to Resta’s gauge-invariant, discrete formulation of the Berry phase [see Eq.(46) of Resta, J. Phys.: Condens. Matter 12 (2000) R107–R143]. However, in Resta’s formula the normalization term doesn’t appear. Could the authors comment on the connection between the two formulas? 6 - As the Berry phase is quantized in this system, the authors should use more meaningful y ticks in Fig. 3(d). In particular, -1 should be indicated. Also, as mentioned above, a different symbol for the invariant should be used. 7 - Below Eq.(9), the statement “manifesting a particle-hole symmetry” is misleading, because particle-hole symmetry is associated with an antiunitary operator, which is not the case here. “Partial chiral symmetry” or “emergent chiral symmetry” is better. 8 - Below that, regarding the statement “it has a topological number”: The authors could be more precise here. I think what they mean is a winding number. Moreover, the winding number is an integer invariant. Therefore, this number should represent the full topological invariant of the model, rather than just a $\mathbb{Z}_2$ factor. Here, the authors could also compare this invariant to their Berry phase invariant. 9 - How does the pseudo-reflection symmetry enter this picture? It seems like I do not need the mirror symmetry to define the winding number and get edge states. What happens if I break mirror symmetry in the overall model? Will I still have the emergent chiral symmetry or are they connected in some way? 10 - The authors provide a nice explanation for the robustness of the topological phases against NN and NNN interaction terms. Based on that, I would expect 4th-neighbor repulsions to have an effect on the entanglement entropy. Have the authors checked that or can the authors comment on how this would affect the topological phases? 11 - In the Appendix, there is a typo: the authors falsely refer to Fig. 3 instead of Fig. 8 (end of page 8, second column). 12 - In the introduction, the authors list references that have demonstrated that mirror symmetry can give rise to novel phases. I would ask the authors to also consider the reference Lau et al., PRB 94, 165164 (2016) in this context.

  • validity: high
  • significance: good
  • originality: high
  • clarity: top
  • formatting: excellent
  • grammar: perfect

Author:  Eytan Grosfeld  on 2022-07-05  [id 2634]

(in reply to Report 2 by Alexander Lau on 2022-04-05)

In this manuscript, the authors study a 1D model of spinless fermions on a zigzag ladder giving rise to topological phases protected by mirror symmetry. They identify a staggered on-site potential to be the important ingredient to obtain topological phases. They determine the ground-state character of the emerging phases as charge-density waves and dimer insulators depending on the filling. The topological nature of the phases is established by Berry phase and entanglement entropy calculations. Moreover, an emergent chiral-like symmetry pins the occuring end states to the center of the gap. The authors further analyze the stability of the topological phases towards interactions using DMRG and QMC. They numerically find that the phases are robust against nearest- and next-nearest neighbor repulsions and provide an intuitive explanation for this finding in terms of the dimer character of the corresponding ground states.

This is an interesting study of a topological model linking non-interacting topology with interaction physics. It is quite timely, in my opinion. The paper has a clear structure and reads nicely. The calculations are sound and the results seem correct overall. The manuscript provides all necessary details to understand the research carried out and the results. I particularly appreciate the intuitive explanations based on the formation of dimers to understand the properties of the system.

We thank the Referee for his constructive comments, and for believing that our work is suitable for publication in Scipost physics.

First of all, the system’s topology is protected by a symmetry that corresponds to a reflection symmetry only in the chain version of the model. In the ladder model, which is the focus of this paper, this symmetry is not a crystalline symmetry, but rather an emergent/artificial symmetry because it doesn’t correspond to an actual mirror plane/line. In other words, the topology in the zigzag ladder is not protected by a real mirror symmetry. This, in turn, renders the protection weaker because the topology is protected “only” by an unphysical symmetry. The authors should be clearer about this and revise their manuscript in this light. In particular, I think the attribute “Mirror-symmetry protected” in the title is misleading in this regard.

In the one dimensional embedding in Fig 1b the mirror symmetry is explicit. We agree that if the zigzag ladder (Fig 1a) is realized in experiment using an optical potential $V(x,y)$, then we need to employ two mirror planes in order to realize the same symmetry. Both lattice models lead otherwise to the same physics, and in practice both versions should be relatively straightforward to realize experimentally. We added a suitable discussion to the paper.

Second, the introduction and discussion of the Berry phase as a topological invariant is problematic. Generally, the Berry phase is not an invariant, because it can assume any value between $0$ and $2\pi$(mod 2). However, it works here because the pseudo-reflection symmetry of the model quantizes the Berry phase to $0$ or $2\pi$ [see Zak, PRL 62, 2747 (1989) or Lau et al., PRB 94, 165164 (2016)]. This is not discussed in the manuscript but merely a numerical observation. Also, even in this case, the Berry phase does not give access to the full $\mathcal{Z}$ topological invariant but only to a $\mathcal{Z}_{2}$ factor of it. This part of the paper should be revised and the quantization be discussed explicitly. Despite my criticism, I think that the paper’s overall results and the type of study are interesting enough to be considered for publication in SciPost Physics after sufficient and satisfactory revision.

We have now provided an elaborated discussion of the topological invariant in the revised manuscript.

$1$ - The authors should be clearer about the considered reflection symmetry and revise their manuscript in this light (see report)

We have now added a discussion about the reflection symmetry to the paper.

$2$ - The part of the paper about the Berry phase invariant should be revised and the quantization be discussed explicitly (see report).

We have revised the manuscript to incorporate more detailed discussion about the Berry phase invariant.

$3$ - After Eq.($2$), I think it would help to define the basis of the Hamiltonian with respect to Fig.$1$ and also indicate the corresponding unit cell in Fig.$1$.

We have shown the unit cells in the zigzag ladder as well as the 1D chain in Fig. 1.

$4$ - The notation $Z$ for the invariant in Eq.(3) is problematic. The symbol $Z$ is used in the literature to denote the set or group of integer numbers. The topological invariant assumes values in this set. Furthermore, as pointed out above, the Berry phase, if quantized, only gives a $Z_{2}$ factor of the full invariant. The authors should rather use a greek or latin character for the invariant.

We have modified the symbols now.

$5$ - Could the authors elaborate more on the origin of Eq.(3)? This formula does not explicitly appear in Ref.34. In particular, I do not see why the normalization term in the denominator is needed. In fact, this formula is otherwise identical to Resta’s gauge-invariant, discrete formulation of the Berry phase [see Eq.(46) of Resta, J. Phys.: Condens. Matter 12 (2000) R107–R143]. However, in Resta’s formula the normalization term doesn’t appear. Could the authors comment on the connection between the two formulas?

The normalization factor does not explicitly appear in Ref $27$ (previously Ref $34$) but rather appears in the definition of the link variable (Eq. $7$ of the Ref $27$). In $1D$ the normalization does not provide anything extra as it will be a real term in the logarithm. We keep the general formula in the manuscript.

$6$ - As the Berry phase is quantized in this system, the authors should use more meaningful $y$ ticks in Fig. $3(d)$. In particular, $-1$ should be indicated. Also, as mentioned above, a different symbol for the invariant should be used.

We have modified the $y$-ticks in Fig. $3(d)$ and used a different symbol for the invariant.

$7$ - Below Eq.$(9)$, the statement “manifesting a particle-hole symmetry” is misleading, because particle-hole symmetry is associated with an antiunitary operator, which is not the case here. “Partial chiral symmetry” or “emergent chiral symmetry” is better.

We incorporated this nomenclature into the paper.

$8$ - Below that, regarding the statement “it has a topological number”: The authors could be more precise here. I think what they mean is a winding number. Moreover, the winding number is an integer invariant. Therefore, this number should represent the full topological invariant of the model, rather than just a $Z_{2}$ factor. Here, the authors could also compare this invariant to their Berry phase invariant.

Yes, this is indeed the same as a winding number for the two-band model. For the two SSH-like projections the maximum number of edge states is $1$ (on each side) in this model, which is consistent with the topological invariant of the full model.

$9$ - How does the pseudo-reflection symmetry enter this picture? It seems like I do not need the mirror symmetry to define the winding number and get edge states. What happens if I break mirror symmetry in the overall model? Will I still have the emergent chiral symmetry or are they connected in some way?

The mirror symmetry in our system is coming from the pattern of the onsite-potential. Moreover, as shown in Sec. III B due to this pattern we can project out the lower or upper two bands and an emergent chiral symmetry develops for the lower and upper bands separately. Thus the emergent chiral symmetry in our system is connected to the mirror symmetry of the overall model. Breaking the mirror symmetry will break the emergent chiral symmetry as well.

$10$ - The authors provide a nice explanation for the robustness of the topological phases against NN and NNN interaction terms. Based on that, I would expect 4th-neighbor repulsions to have an effect on the entanglement entropy. Have the authors checked that or can the authors comment on how this would affect the topological phases?

We thank the referee for this comment. Indeed, we have now verified that longer range interactions will have an effect on the entanglement entropy. Since the dimers are formed at every fourth NN bond, the presence of a third neighbor repulsion will affect the formation of dimers and hence the robustness of topological phase will be altered. This is now discussed in the paper and we added a new graph demonstrating the transition to a charge-density-wave for strong enough NNNN interactions.

$11$ - In the Appendix, there is a typo: the authors falsely refer to Fig. $3$ instead of Fig. $8$ (end of page $8$, second column).

We have corrected it in the revised manuscript.

$12$ - In the introduction, the authors list references that have demonstrated that mirror symmetry can give rise to novel phases. I would ask the authors to also consider the reference Lau et al., PRB 94, 165164 (2016) in this context.

We thank the referee for bringing this interesting work to our notice. We have now added this work in our introduction.

---

## Round 1 · Referee Report · Anonymous (Referee 3) · 2022-4-6

Weaknesses

  1. Clear motivation missing
  2. Weak conclusions

Report

The authors study the topology of a one-dimensional insulator with mirror symmetry. The only motivation that I find for this work is summarized by the statement that "the investigation of topological crystalline phases and their robustness against various perturbations and many-body interactions is in a nascent stage". In reality, the classification of the topological crystalline phases (both non-interacting and many-body) is well understood, here I list some of the relevant works:

PRB,96,205106, 2017
PRX,8, 011040, 2018
physica status solidi (b) 258 (1), 2000090, 2021
arXiv:1810.00801

The system that the authors study belongs to the class of "symmetry obstructed atomic insulators", such insulators do not have topologically protected end-states. In absence of chiral (anti)symmetry, the end states that are found for 3/4 (W negative) and 1/4 filling (W positive) can be easily moved outside of the gap by an onsite potential applied to the ends of the chain. In other words, the emergent chiral symmetry that authors find for the lower (upper) two bands can be easily broken.

I fail to understand what is the advantage of using entanglement entropy S to quantify topology in this system. Since the computation needs to be performed with OBC, such a calculation (unlike Eq. 3) does not give "bulk" topological invariant. In fact, S is not quantized to the value of "ln 2" in the topological phase as seen in Fig. 4a (because the system size is too small for certain W). The only thing S detects is that with OBC there is ground-state degeneracy for large systems, but this is only true as long as the two end states are in the gap. One possibility for defining the bulk topological invariant for the interacting case is to replace Eq. 3 with "Resta-like expression" from the modern theory of electrical polarization (see also Phys. Rev. Lett. 118, 216402).
  • validity: ok
  • significance: low
  • originality: low
  • clarity: good
  • formatting: good
  • grammar: excellent

Author:  Eytan Grosfeld  on 2022-07-05  [id 2633]

(in reply to Report 3 on 2022-04-06)

We thank the Referee for reading our work and for his/her comments.

The authors study the topology of a one-dimensional insulator with mirror symmetry. The only motivation that I find for this work is summarized by the statement that ”the investigation of topological crystalline phases and their robustness against various perturbations and many-body interactions is in a nascent stage”. In reality, the classification of the topological crystalline phases (both non-interacting and many-body) is well understood, here I list some of the relevant works: PRB,96,205106, 2017 PRX,8, 011040, 2018 physica status solidi (b) 258 (1), 2000090, 2021 arXiv:1810.00801

The mentioned works report the classification of topological crystalline insulators in higher dimensions. However, a thorough study of the role of many-body interactions and the resulting phase transitions is still lacking partly due to inability to access larger system sizes. Our work provides a simple one-dimensional model with mirror symmetry where we can straightforwardly exact the role of interactions by going to larger system sizes. In the revised manuscript, we have now added another interesting finding per the suggestion of referee 2, namely an interaction-induced topological-phase-transition from a topological non-trivial dimer insulating phase to a topologically trivial charge density wave phase on varying the NNNN interaction strength. We have also made the motivation clearer in the revised manuscript: indeed, our work sheds light on the role of multi-range interactions and their effect on phase transitions that occur in mirror-symmetry protected phases, whose details and nature (topological, non-topological) are not captured by classifications alone.

The system that the authors study belongs to the class of "symmetry obstructed atomic insulators", such insulators do not have topologically protected end-states. In absence of chiral (anti)symmetry, the end states that are found for 3/4 (W negative) and 1/4 filling (W positive) can be easily moved outside of the gap by an onsite potential applied to the ends of the chain. In other words, the emergent chiral symmetry that authors find for the lower (upper) two bands can be easily broken.

In the absence of chiral symmetry, the end-modes can be moved outside of the gap by applying the onsite potential at the ends. However, this is a special type of perturbation and does not change the bulk topology. It has been already established (Lau et al., PRB 94, 165164 (2016)) that the bulk-boundary correspondence in mirror-symmetry protected topological systems is very ``unconventional". We do not claim here that the edge states are protected by the emergent chiral symmetry against any kind of perturbation; instead we claim that the emergent chiral symmetry protects them against next-neighbor hopping, for which we derived the projection. This serves to explain the form of the non-interacting spectrum appearing in Fig. 5a, as well as to supply a method for analyically calculating the topological number via a winding number of two bands, which is otherwise complicated in a four-band model. In practice, controlling the energies of the two endmost sites is an experimental necessity also in order to satisfy the requirement of a chiral symmetry in the SSH chain; the latter was nevertheless realized experimentally in various works, so we do not foresee any problems also with realizing our model, which comprises two intertwined SSH chains that together break chiral symmetry but keep mirror-symmetry intact. The projected individual chains each have an effective chiral symmetry protecting them against certain bulk perturbations.

I fail to understand what is the advantage of using entanglement entropy S to quantify topology in this system. Since the computation needs to be performed with OBC, such a calculation (unlike Eq. 3) does not give "bulk" topological invariant. In fact, $S$ is not quantized to the value of "$ln 2$" in the topological phase as seen in Fig. 4a (because the system size is too small for certain $W$). The only thing $S$ detects is that with OBC there is ground-state degeneracy for large systems, but this is only true as long as the two end states are in the gap. One possibility for defining the bulk topological invariant for the interacting case is to replace Eq. $3$ with "Resta-like expression" from the modern theory of electrical polarization (see also Phys. Rev. Lett. 118, 216402)

In the revised manuscript, we have now characterized the topology in the interacting case by calculating the topological invariant using exact diagonalization with twisted boundary conditions. This calculation confirms the results discussed using the entanglement entropy. For the case of OBC, a careful study of the entanglement entropy (sensitive to the bulk dimers and edge state degeneracy) as well as the change in the entanglement entropy under the addition of a single particle (sensitive to the edge state degeneracy) contain telltale signs of phase transitions and topology so they are suitable for our study here: either by not observing any change when monitoring the entanglement as we increase the interaction strength from zero (Fig. 7), or by seeing signs of a phase transition as in the new Fig. 8. In parallel to using the entanglement entropy to characterize the newly discussed topological-phase-transition induced by NNNN interactions, we also performed QMC studies to demonstrate the change in the bulk order parameter. Moreover, in Fig. 4a within the topological phases the entanglement entropy does get quantized at all values of the onsite-potential except at $W=0$ (which corresponds to a gapless phase) and inside a narrow window close to it; this is a consequence of numerical errors due to system size effect and can be resolved by increasing the system size further.

---

## Round 2 · Referee Report · Alexander Lau (Referee 2) · 2022-7-20

Strengths

1 - interesting study of a topological model linking non-interacting topology with interaction physics
2 - the paper has a clear structure and reads nicely
3 - the calculations are sound and the results seem correct overall
4 - the manuscript provides all necessary details to understand the research carried out and the results
5 - provides an intuitive picture based on the formation of dimers to understand the numerical results, in particular the robustness of the topological phases in the presence of interactions
6 - finds a transition from a topological dimer insulator to a trivial charge-density-wave insulator for longer-range interactions

Weaknesses

-

Report

The authors have responded to all of my concerns and suggestions and have incorporated them in the manuscript. In particular, my concerns regarding the mirror symmetry and the introduction of the topological invariant have been clarified sufficiently. One of my comments has also motivated the authors to add a new and interesting aspect to their work, namely that longer-range hopping (NNNN) leads to a phase transition from the topological dimer state to a trivial charge-density-wave state. They have also shifted the focus of their work to this aspect, which strengthens the manuscript, in my opinion.

I am satisfied with the revised manuscript and believe that it satisfies all general acceptance criteria of SciPost Physics. Moreover, I also believe that this work has the potential to trigger more follow-up work exploring similar interaction effects and phase transitions involving other crystalline topological phases (expectation criterion no. 3). I therefore recommend publication in SciPost Physics.

Requested changes

1 - Typo in the new paragraph at the end of Appendix A: “Concomitantly, the structure factor S(π/2) increases from a value close to 0.3125 to a value close to 0.0625.” From Fig. 10 and the text, it looks as if the V3=0 value should rather be 0.03125. The same typo is repeated a bit later in the same paragraph.

---

## Round 2 · Referee Report · Anonymous (Referee 3) · 2022-7-22

Weaknesses

Clear motivation missing

Report

Even in the revised version, I do not see the motivation to study the details of a toy model. In my view, it makes sense to study a toy model if it is representative of a larger class of models (corresponding to the given topological phase). Unfortunately, this work does not offer any new results that apply to the whole class of mirror-symmetric topological phases.

---

## Round 2 · Author Response

Dear Editor,

We would like to thank you and the Referees for their positive evaluation and suggestions.

We want to highlight that our work provides a simple one-dimensional model that admits a topological phase that is protected by time-reversal symmetry and mirror-symmetry alone, amenable to exploration in the presence of many-body interactions. In the revised manuscript, we have demonstrated an interaction-induced transition from a topological mirror-symmetry protected dimer-insulating phase to a topologically trivial charge density wave phase. To the best of our knowledge, this is the first time that such an interaction-induced topological phase transition is shown between a fermionic 1d crystalline topological phase and a charge-density wave. Thus we feel that the work is an important contribution to the scientific community and has the scope for greater exploration in the future, for example with other lattice symmetries or with additional types of interactions.

We have revised our manuscript according to the suggestions of the referees. We also wanted to mention that symmetry-protected topological phases in one dimension are sometimes referred to as symmetry-obstructed atomic insulators as pointed out by Referee 1 and 3. While we prefer to go with the also prevalent nomenclature of topological insulators, we have added a paragraph to the discussion that highlights this point. Below we provide a point-by-point response to the questions raised by the Referees.

We thank you for your consideration and look forward to hearing back from you.

Yours sincerely,
Devendra Singh Bhakuni, Amrita Ghosh, Eytan Grosfeld

---

## Round 2 · List of Changes

Major changes:

  1. We modified the abstract, the introduction and conclusions to highlight the motivation for the paper in terms of the effect of interactions on mirror-symmetry-protected 1d topological phases. In particular, we added a treatment of NNNN interactions which can drive a phase transition at critical strength. In light of that, we also changed the title of the paper to reflect the stronger focus on interactions.

  2. In the new Fig 7c and 7d we report the topological invariant in interacting cases.

  3. In Fig. 8 and in the related discussion on pages 6-7 we demonstrate a phase transition that occurs at a critical strength of NNNN interactions using the entanglement entropy. New figures and additional discussion in the appendix describe the changes in the order parameters as a function of the strength of NNNN interactions.

---

## Editorial Decision

published